# Tubular Endogenous Erythropoietin Protects Renal Function against Ischemic Reperfusion Injury

**DOI:** 10.3390/ijms25021223

**Published:** 2024-01-19

**Authors:** Yukiko Yasuoka, Yuichiro Izumi, Takashi Fukuyama, Tomomi Oshima, Taiga Yamazaki, Takayuki Uematsu, Noritada Kobayashi, Masayoshi Nanami, Yoshitaka Shimada, Yasushi Nagaba, Masashi Mukoyama, Jeff M. Sands, Noriko Takahashi, Katsumasa Kawahara, Hiroshi Nonoguchi

**Affiliations:** 1Department of Physiology, Kitasato University School of Medicine, 1-15-1 Kitasato, Minami-ku, Sagamihara 252-0374, Kanagawa, Japan; yasuoka@med.kitasato-u.ac.jp (Y.Y.); tomomio@kitasato-u.ac.jp (T.O.); kawahara@kitasato-u.ac.jp (K.K.); 2Department of Nephrology, Kumamoto University Graduate School of Medical Sciences, 1-1-1 Honjo, Chuo-ku, Kumamoto 860-8556, Kumamoto, Japan; izumi_yu@kumamoto-u.ac.jp (Y.I.); mmuko@kumamoto-u.ac.jp (M.M.); 3Division of Biomedical Research, Kitasato University Medical Center, 6-100 Arai, Kitamoto 364-8501, Saitama, Japan; fukuyam@insti.kitasato-u.ac.jp (T.F.); tyamazak@insti.kitasato-u.ac.jp (T.Y.); tuematsu@insti.kitasato-u.ac.jp (T.U.); kenchu@insti.kitasato-u.ac.jp (N.K.); 4Division of Kidney and Dialysis, Department of Internal Medicine, Hyogo Medical University, 1-1 Mukogawa-cho, Nishinomiya 663-8501, Hyogo, Japan; m-nanami@hyo-med.ac.jp; 5Division of Internal Medicine, Kitasato University Medical Center, 6-100 Arai, Kitamoto 364-8501, Saitama, Japan; yoshi@insti.kitasato-u.ac.jp (Y.S.); nagaba-y@insti.kitasato-u.ac.jp (Y.N.); 6Renal Division, Department of Medicine, Emory University School of Medicine, 1639 Pierce Drive, WMB Room 3313, Atlanta, GA 30322, USA; jeff.sands@emory.edu

**Keywords:** erythropoietin, PHD inhibitor, hypoxia, deglycosylation, proximal tubules, distal tubules, renal erythropoietin-producing interstitial cells

## Abstract

Many large-scale studies show that exogenous erythropoietin, erythropoiesis-stimulating agents, lack any renoprotective effects. We investigated the effects of endogenous erythropoietin on renal function in kidney ischemic reperfusion injury (IRI) using the prolyl hydroxylase domain (PHD) inhibitor, Roxadustat (ROX). Four h of hypoxia (7% O_2_) and 4 h treatment by ROX prior to IRI did not improve renal function. In contrast, 24–72 h pretreatment by ROX significantly improved the decline of renal function caused by IRI. Hypoxia and 4 h ROX increased interstitial cells-derived Epo production by 75- and 6-fold, respectively, before IRI, and worked similarly to exogenous Epo. ROX treatment for 24–72 h increased Epo production during IRI by 9-fold. Immunohistochemistry revealed that 24 h ROX treatment induced Epo production in proximal and distal tubules and worked similarly to endogenous Epo. Our data show that tubular endogenous Epo production induced by 24–72 h ROX treatment results in renoprotection but peritubular exogenous Epo production by interstitial cells induced by hypoxia and 4 h ROX treatment did not. Stimulation of tubular, but not peritubular, Epo production may link to renoprotection.

## 1. Introduction

Erythropoietin (Epo) is found in urine of anemic patients [1,2]. Erythropoiesis-stimulating agents (ESAs) changed the treatment of renal anemia [3,4,5,6]. Many large-scale studies investigated the effects of ESAs for the progression of chronic renal failure and found that ESAs have no renoprotective effects [5,7,8,9]. Prolyl hydroxylase domain (PHD) inhibitors have been used for the treatment of renal anemia instead of ESAs due to the ease of oral intake [6,10,11,12,13,14,15]. Epo produced by ESAs works similarly to exogenous Epo. In contrast, PHD inhibitors stimulate endogenous Epo production. PHD inhibitors inhibit PHD activity and increase HIF2α expression to increase endogenous Epo production [16,17,18]. Production of endogenous Epo has additional effects beyond Epo itself. Endogenous Epo is known to decrease interleukin 6 and monocyte chemoattractant protein-1 production by macrophages [19]. The difference in the physiological effect between exogenous and endogenous Epo is not clear yet. Epo pretreatment has been used as a substitute for preconditioning to ischemia [20,21,22,23,24,25,26]. Some reports show a significant improvement of ischemic reperfusion injury (IRI) by Epo pretreatment due to the reduction in inflammation or oxidative stress, but some did not. Inflammation is known to affect anemia [27]. Epo pretreatment examined the effects of exogenous Epo on renal function. We have invented the deglycosylation-coupled Western blotting method for Epo in blood, urine, and tissue [28,29,30]. Our method improved the sensitivity and specificity of Epo detection, especially in tissue. We also investigated tubular Epo production by immunohistochemistry (IHC) [28,31,32,33]. Our IHC helps to distinguish the site of Epo production and exogenous/endogenous Epo.

There are two sites of Epo production in the kidney: interstitial cells and nephrons [28,31,32,33,34,35,36,37,38,39]. Epo production by the interstitial cells is not seen in control conditions and is stimulated to extreme levels only in severe hypoxia and anemia [16,40,41]. In contrast, Epo production by the nephron is observed in the control condition and is stimulated several fold by activation of the renin–angiotensin–aldosterone system (RAS) [42,43,44,45,46]. In our study, exogenous Epo means peritubular production of Epo by the interstitial cells. Epo is also produced by the nephron and acts similarly to endogenous Epo. Thus, exogenous Epo means peritubular Epo and endogenous Epo is tubular Epo in our study. We investigated the effects of endogenous Epo on renal function in IRI, a model of acute kidney injury (AKI).

## 2. Results

### 2.1. Serum Creatinine Levels Pre- and Post-IRI

IRI caused an increase in serum creatinine from 0.53 ± 0.02 to 2.52 ± 0.29 mg/dL, the same as previous reports [21,22,23,24,25,26]. Hematoxylin–eosin (HE) staining of the kidney showed damage to proximal tubules but not in the glomerulus (Appendix A). Severe hypoxia and 4 h ROX (R4) treatment did not decrease the serum creatinine level. In contrast, 24 h ROX (R24) and 72 h ROX (R72) significantly decreased creatinine levels (2.52 ± 0.29, 1.69 ± 0.64, 1.40 ± 0.26, 0.76 ± 0.05 **, and 0.97 ± 0.19 ** mg/dL by control, hypoxia, R4, R24, and R72, respectively, ** *p* < 0.01 by ANOVA and Scheffe’s multiple comparison, Figure 1).

### 2.2. Plasma Epo Concentration before and after IRI

Plasma Epo concentration was examined before and after IRI. A high plasma Epo concentration was observed in the hypoxia and R4 groups (Figure 2). Only the R4 group showed a high plasma Epo concentration after IRI. This high plasma Epo concentration in the R4 group was thought to be caused by Epo production before IRI.

### 2.3. Western Blot Analysis of Epo Production during IRI

To reveal the amount of Epo production during IRI, Western blot analysis was performed using the kidney after IRI. We first examined Epo detection both in the absence and presence of Peptide-N-Glycosidase F (PNGase) (Figure 3a,b). It is difficult to detect glycosylated Epo at 35–38 kDa (left lanes in Figure 3a,b). It is easy to detect deglycosylated Epo at 22 kDa (right lanes in Figure 3a–c). Deglycosylated Epo production was examined at pre- and post-IRI. Hypoxia caused high Epo production before IRI and R4 showed a slight increase in Epo production (75− and 6−fold increase compared with control, Figure 3c). Epo production by the R4 group is lower than in our previous report. Our previous reports examined Epo production at 6 h after the injection of ROX. Therefore, the maximum increase in Epo production was thought to be caused during the formation of IRI in the R4 group. In contrast, Epo production after IRI was highest in R24 group (9-fold increase compared with control, Figure 3c).

### 2.4. Epo mRNA Expression before and after IRI

*Epo* mRNA expression was very high in the hypoxia and R4 groups and very low in the R24 and R72 groups before IRI (Figure 4). *Epo* mRNA after IRI was highest in the R4 group and very low in the hypoxia group. The R24 and R72 groups showed slight increases in *Epo* mRNA expression.

### 2.5. Immunohistochemistry (IHC) of Epo Production by the Kidney

Since Epo production after IRI in the kidney was stimulated by R24, we examined the site of Epo production by IHC. The R24 group showed production of Epo by the proximal convoluted tubules and the distal tubules (brown staining in Figure 5a). Proximal tubules showed higher production than distal tubules (Figure 5b,c). Interstitial cells showed no production of Epo by R24.

### 2.6. Western Blot Analysis of Autophagy and Apoptosis

We examined the expression of LC3-II and p62 in the kidney to determine whether autophagy is involved in the renoprotective effects of Epo. LC3-II and p62 expressions were not changed by hypoxia and ROX (Figure 6a). Next, BAX and Bcl-2 expressions in the kidney were examined to determine the participation of apoptosis in the renoprotective effects of R24. Bcl-2 expression was lower in the kidney cortex than medulla. BAX and Bcl-2 expressions were not changed by hypoxia or ROX (Figure 6b).

### 2.7. Tubular Protein and mRNA Expressions in the Kidney during IRI

The expressions of *Hif2α* and *Hif1α* during IRI were not changed by hypoxia and ROX except for the decline of *Phd2* mRNA expression in R24-72 (Table 1). PEPCK (Pck-1) is mainly present in the proximal convoluted tubules. We examined phosphoenolpyruvate carboxykinase (PEPCK) expression after IRI. R24-72 showed higher expression than hypoxia and R4 groups (Figure 7a). *Gr* (*Nr3c1*) is present in proximal convoluted tubules and early distal tubules. *Rhcg* is present in distal convoluted tubules and collecting ducts. The mRNA expression of *Gr* and *Rhcg* were increased in the R24 group (Figure 7b).

## 3. Discussion

Our present study showed that administration of ROX 24–72 h prior to the induction of IRI protected against the decline of renal function by stimulating tubular endogenous Epo production. Stimulation of Epo production before IRI by 4 h hypoxia and during IRI by R4 largely increased interstitial cells-derived Epo production and served as exogenous Epo but did not protect against the decline of renal function. In our IRI model, hypoxia and R4 stimulated Epo production by the interstitial cells judging from the high plasma Epo concentration. Epo production by the nephron during IRI by R24-72 was small and did not increase the plasma level. Our deglycosylated Western blot and immunohistochemistry studies revealed Epo production during IRI.

Renoprotective effects of PHD inhibitors have been reported but the results are controversial [21,22,23,24,25,26,47,48,49,50,51]. Many mechanisms such as the reduction in inflammation, oxidative stress, or apoptosis have been suggested. We have employed deglycosylation-coupled Western blotting and immunohistochemistry for the detection of Epo protein in the kidney. We found that exogenous Epo has no renoprotective effects as previously revealed by many large-scale studies [7,8]. The *Hir2α* and *Hif1α* during IRI was not changed by four groups except for the decline of *Phd2* mRNA expression in R24-72, suggesting that R24-72 caused endogenous Epo production by some parts of proximal and distal tubules. IRI is known to induce damage to proximal tubules. The increased expression of PEPCK and *Gr* mRNA by R24-72 revealed the recovery of proximal tubules. The increase in *Rhcg* mRNA expression showed the recovery of collecting ducts. Therefore, R24-72 seemed to recover not only proximal but also distal tubular function.

We investigated the mechanisms of renoprotection by Epo production. Autophagy and reduction in apoptosis can induce renoprotection [52,53]. However, Western blot analysis of LC3-II and p62 expression was not changed by 24–72 h ROX. The expression of BAX and Bcl-2 were also unchanged. These data suggest the lack of autophagy and apoptosis induced by ROX.

The Epo receptor has some role in Epo effects on tubular function [53]. Epo receptors in the kidney exist in the inner medullary collecting ducts [54]. Inner medullary collecting ducts are the terminal portion of the nephron and are an important site for urine concentration. Therefore, the renoprotective effect of tubule-produced Epo is not thought to be caused by activation of the Epo receptor.

It is interesting that tubules are the site of Epo production in this IRI model. Renal Epo producing (REP) cells have been known to produce large amounts of Epo, but a small production of Epo by the nephron was enough to protect against renal failure by IRI. We and others have shown that the RAS regulates Epo production by the collecting duct [31,33,40,42,43,44,45,46]. Taken together, endogenous Epo production by the tubules is thought to play an important role in the maintenance of kidney function, even with the lower capacity for Epo production. We have shown that aldosterone and angiotensin-II stimulate Epo production by the collecting ducts, especially by the intercalated cells [31,33]. Endogenous Epo production by the proximal tubules is larger than by the distal tubules in this IRI model. This is probably because IRI causes a larger amount of damage to the proximal tubules than distal ones, suggesting that proximal and distal tubules can produce Epo in response to local hypoxia. We have distinguished peritubular Epo production by REP cells from tubular endogenous Epo production. Our data clearly show that a small amount of tubular endogenous Epo production has a renoprotective effect, while the large amount of REP cell-derived peritubular Epo production has no effect on tubular function. Large-scale studies showed the lack of renoprotective effect by ESAs, the same as with 4 h hypoxia and R4 in our study [5,7,8,9]. Considering the renoprotection caused by Epo production, methods to induce tubular Epo production would be useful. One of the keys to finding useful tools for renoprotection is that the RAS has some interaction with tubular Epo production [31,33,40,42,43,44,45,46]. Our model is close to the situation of kidney transplantation. The use of PHD inhibitors 24 h before the transplant may largely decrease the damage that occurs after transplantation.

In summary, 24–72 h ROX caused renoproctive effects by stimulating tubular endogenous Epo production, but peritubular interstitial cells-derived exogenous Epo production induced by 4 h hypoxia and 4 h ROX did not repair tubular dysfunction. Methods to induce tubular endogenous Epo production might have a crucial role for renoprotection.

## 4. Materials and Methods

### 4.1. Materials and Animals

Male Sprague Dawley rats (200–250 g, Japan SLC, Hamamatsu, Japan) were used in our study. The rats were randomly assigned to two groups with the absence and presence of IRI: w/o IRI group; control and ROX group; w/IRI group were divided into 5 groups, control, 4 h hypoxia, 4 h ROX, 24 h ROX, and 72 h ROX groups. ROX (FG-4592; MedChemExpress, Monmouth Junction, NJ, USA) were dissolved in 5% glucose with 32.5 mmol/L NaOH. Rats in the ROX and ROX–IRI groups were pretreated intraperitoneal injection of ROX (50 mg/kg body weight). Control and IRI groups were injected with the same dose of 5% glucose with 32.5 mmol/L NaOH. Hypoxia group: rats were exposed to 7% O_2_ and 93% N_2_ for 4 h. After 4, 24, or 72 h pretreatment by ROX or 4 h hypoxia, all rats were anesthetized with 1.5% isoflurane in 30% O_2_ (a mixture of 100% O_2_ and air), and the left kidney was removed. IRI and ROX-IRI groups: right renal pedicle was clamped using non-traumatic vascular clips for 45 min and, then, the clamps were released to allow reperfusion to the kidney. Control and ROX groups: rats underwent the same procedure without clamping of the renal pedicles. After 24 h of I/R, all rats were anesthetized and blood samples were collected from the abdominal aorta. Kidneys were fixed in 4% PFA, embedded in paraffin, and stored in liquid nitrogen. Our protocols were checked and approved by the Ethics Committee at Kitasato University Medical Center (H24-011 and 2022-12) and Kitasato University School of Medicine (2022-140 and 2023-077).

The hypoxia group was expected to increase plasma Epo concentration at the starting time of the IRI operation and the R4 group was expected to produce Epo during the procedure of IRI [28,32]. We have shown that ROX caused high plasma Epo levels 6 h after administration [32]. The 24 h and 72 h ROX groups were expected to increase Epo production after the IRI operation.

### 4.2. Serum Creatinine and Plasma Epo Concentration Measurements

Blood samples were centrifuged at 3000× *g* for 10 min. Supernatants were collected to measure serum creatinine levels using StatSensor (Nova Biomedical, Waltham, MA, USA). Plasma Epo concentrations were measured by CLEIA (SRL, Tokyo, Japan).

### 4.3. Western Blot Analysis

Deglycosylation-coupled Western blot analysis was performed as described previously [26,31,32]. Protein was extracted from the kidney using CelLytic MT (C-3228; Sigma-Aldrich, Burlington, MA, USA) plus protease inhibitor (05892970001, Roche, Basel, Switzerland) and was used for Western blotting. Samples were deglycosylated using Peptide-N-Glycosidase F (PNGase, 4450; Takara Bio, Kusatsu, Japan). A total of 1 μL of 10% SDS was added to 10 μL samples and boiled for 3 min. Then, 11 μL of 2× stabilizing buffer was added. After the addition of 1 μL of PBS (to measure glycosylated Epo) or PNGase (to measure deglycosylated Epo), samples were incubated in a water bath for 17–20 h at 37 °C. After the incubation, samples were spun down and the supernatant was used for SDS-PAGE (10–20% gradient gel, 414893; Cosmo Bio, Tokyo, Japan). The 2× stabilizing buffer contained 125 mM Tris-HCl (pH 8.6), 48 mM EDTA, 4% Nonidet P-40 and 8% 2-mercaptoethanol. Recombinant rat Epo (rRatEpo, 592302; BioLegend, San Diego, USA) was used as a positive control both in glycosylated and deglycosylated samples. After SDS-PAGE, proteins were transferred to a PVDF membrane (Immobilon-P, IPVH00010; Merck Millipore, Burlington, MA, USA) for 60–90 min at 120 mA. The membrane was blocked with 5% skim milk (Morinaga, Tokyo, Japan) for 60 min and incubated with the antibody against Epo (sc-5290, 1:500; Santa Cruz) for 60 min at room temperature. After washing, the membrane was incubated with a secondary antibody (goat anti-mouse IgG (H+L) (115-035-166, 1:5000; Jackson ImmunoResearch Laboratories, West Grove, PA, USA) for 60 min. Bands were visualized by the ECL Select Western Blotting Detection System (RPN2235; GE Healthcare Bio-Science AB, Uppsala, Sweden) and LAS 4000 (Fujifilm, Tokyo, Japan). After measuring Epo protein expression, the membrane was stripped (stripping solution, Wako, RR39LR, Tokyo, Japan) and reprobed with the antibody against β-actin (MBL, M177-3, Tokyo, Japan), GAPDH (Santa Cruz, sc-32233) or α-tubulin (Santa Cruz, sc-69969) for the normalization of the band. Western blotting of LC3-II (Cell Signaling, D3U4C) [55], p62 (MBL, PM066), Bcl-2 (Santa Cruz, sc-23960), BAX (GeneTex, 127309), and PEPCK (Cayman, 10004943) was also performed in the same way as for Epo.

### 4.4. Real-Time Quantitative RT-PCR

RNA was extracted from kidney and liver using Qiacube and the RNeasy Mini Kit (74106; Quiagen, Venlo, The Netherlands) as described previously [28,33]. cDNA was synthesized using a Takara PrimeScript II 1st strand cDNA Synthesis Kit (6210; Takara Bio). Real-time PCR was performed using probes form Applied Biosystems, Waltham, MA, USA (*β-actin* Rn00667869_m1, *Epo* Rn00667869_m1, *Hif2α* Rn00576515_m1, *Hif1α* Rn01472831_m1, *Phd2* Rn00710295_m1, *Gr* (*Nr3c1*) Rn00561369_m1, *Rhcg* Rn00788284_m1), and Premix Ex Taq (RP39LR; Takara Bio). mRNA expressions in control and IRI rats were compared by relative gene expression data using real-time quantitative PCR and the 2^−ΔΔCT^.

### 4.5. Immunohistochemistry

Kidney sections were immuno-stained as described previously [28,31,32,33]. In brief, the sections were blocked with 5% normal goat serum and reacted with rabbit polyclonal anti-human Epo antibody (sc-7956, 1:10; Santa Cruz Biotechnology, Santa Cruz, CA, USA), followed by Histofine simple stain MAX-PO (414341F; Nichirei Bioscience, Tokyo, Japan). Sections were stained using DAB liquid system (BSB 0016; Bio SB, Santa Barbara, CA, USA) and counterstained with Mayer’s haematoxylin (30002; Muto Pure Chemicals, Tokyo, Japan).

Images were obtained using an optical microscope (Axio Imager M2; Carl Zeiss, Oberkochen, Germany) with a digital camera (AxioCam 506, Carl Zeiss). Captured images were analysed using an image analysing system (ZEN 2, Carl Zeiss).

### 4.6. Statistical Analyses

Data are expressed as mean ± SEM. Statistical significance was performed using Excel Statics (BellCurve, Tokyo, Japan). Statistical significance was analyzed using ANOVA and multiple comparison of Scheffe. *p* < 0.05 was considered statistically significant.

## Figures and Tables

**Figure 1 ijms-25-01223-f001:**
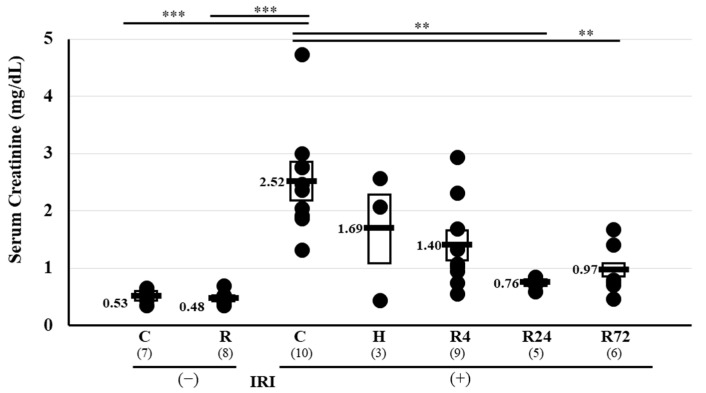
Changes in renal function by IRI. Serum creatinine levels by IRI and Epo stimulation. IRI caused an increase in serum creatinine from 0.53 to 2.52 mg/dL. Hypoxia and R4 did not decrease serum creatinine levels. In contrast, R24 and R72 decreased serum creatinine levels to 0.76 and 0.97, respectively. ** *p* < 0.01, *** *p* < 0.001 by ANOVA and Scheffe’s multiple comparison. C, control; H, hypoxia; R, ROX.

**Figure 2 ijms-25-01223-f002:**
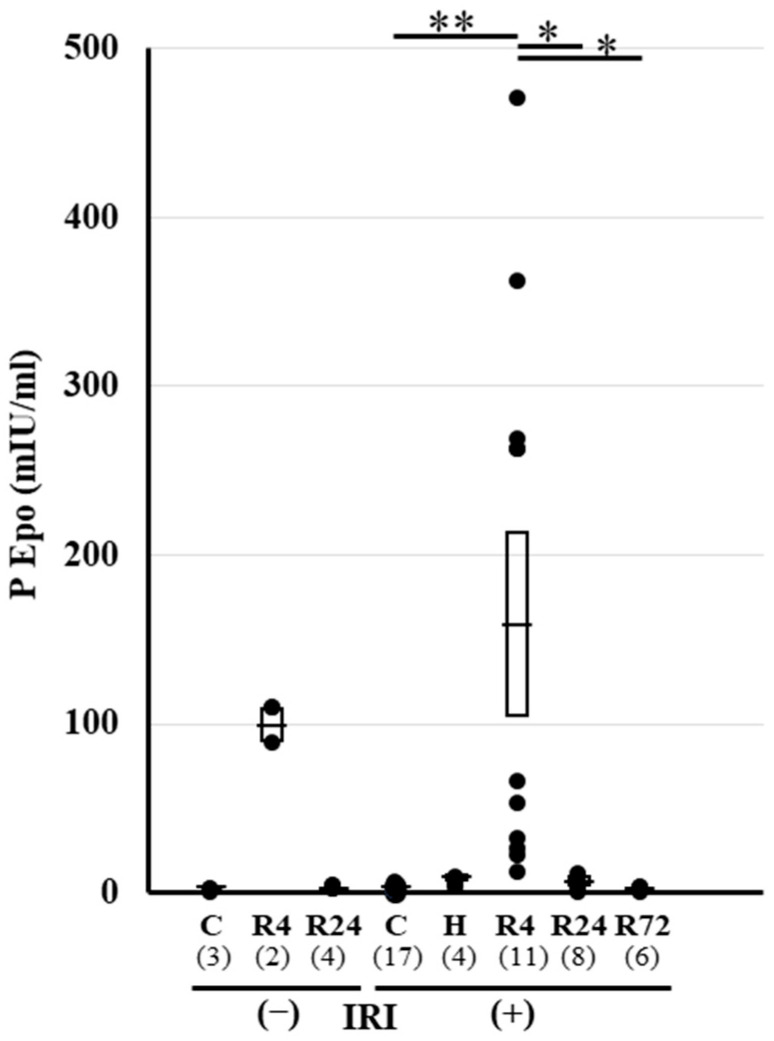
Plasma Epo levels after IRI. Plasma levels of Epo in each group. Hypoxia caused a large increase in plasma Epo level. R4 caused a slight increase in plasma Epo level. After IRI, R4 caused an increase in plasma Epo level but hypoxia and R24−72 did not change the levels. The numbers in parenthesis shows the number of samples. * *p* < 0.05, ** *p* < 0.01 by ANOVA and Scheffe’s multiple comparison. C, control; H, hypoxia; R, ROX.

**Figure 3 ijms-25-01223-f003:**
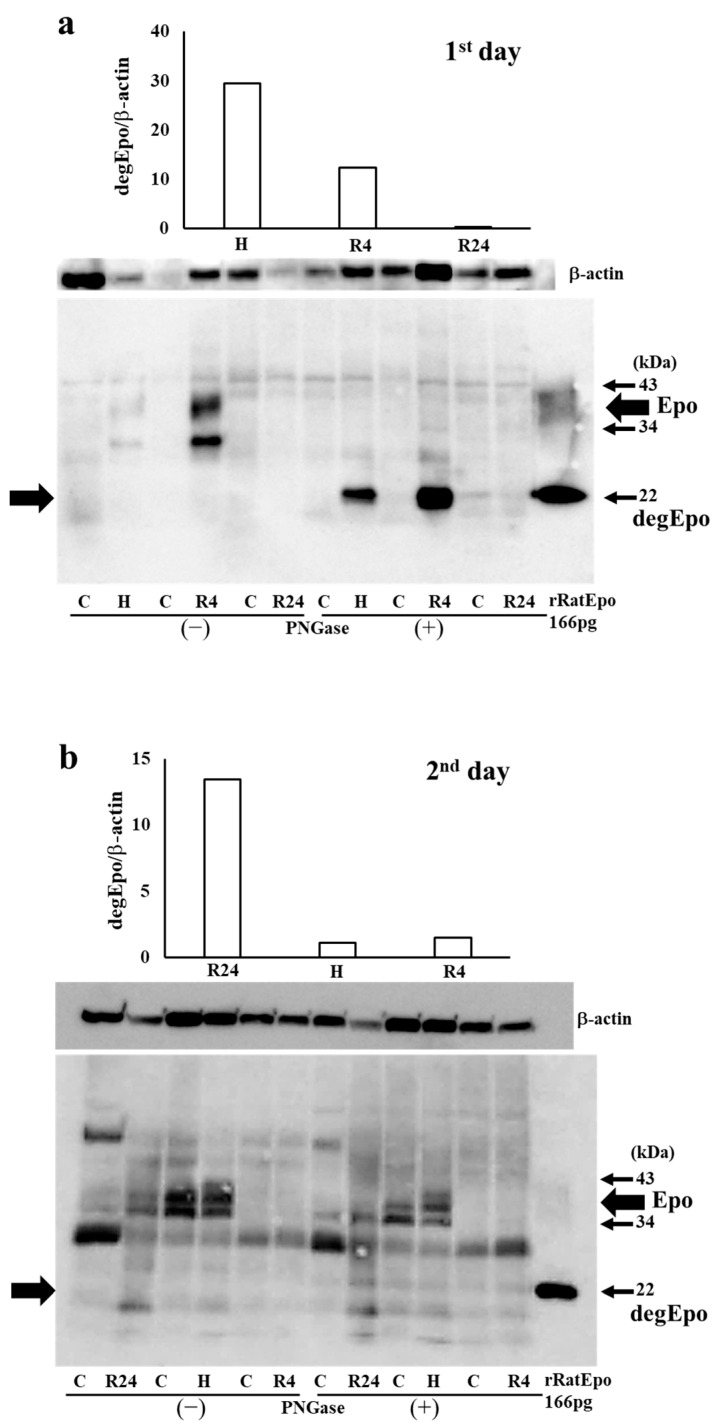
(**a**). Western blot analysis of Epo protein before IRI in the absence and presence of PNGase. Epo production by the kidney before IRI. A typical gel of pre-IRI is shown. Highest production is observed by 4 h hypoxia. C, control; H, hypoxia; R, ROX. (**b**). Western blot analysis of Epo protein after IRI in the absence and presence of PNGase. Epo production by the kidney after IRI. A typical gel of post-IRI is shown. Highest production is observed by 24 h ROX. C, control; H, hypoxia; R, ROX. (**c**). Epo production by the kidney before and after IRI. Western blot analysis of Epo production by the renal cortex. Hypoxia induced a large amount of Epo production before IRI. After IRI, R24 caused a slight increase in Epo production. R4 did not induce Epo production after IRI, suggesting that the high plasma Epo level is caused by Epo production before IRI. Deglycosylated Epo expression at 22 kDa (degEpo shown by the left arrow head) was corrected by β-actin expression (degEpo/β-actin), * *p* < 0.05 by ANOVA and Scheffe’s multiple comparison. n = 4–5. C, control; H, hypoxia; R, ROX.

**Figure 4 ijms-25-01223-f004:**
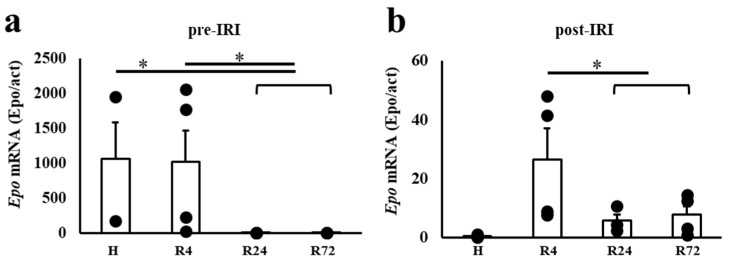
*Epo* mRNA expression before and after IRI. (**a**): Hypoxia and R4 induced large increases in *Epo* mRNA expressions before IRI. (**b**): A slight stimulation of *Epo* mRNA expression was observed by R4, R24, and R72 after IRI. n = 3–7. H, hypoxia; R, ROX. * *p* < 0.05 by ANOVA and Dunnett’s multiple comparison.

**Figure 5 ijms-25-01223-f005:**
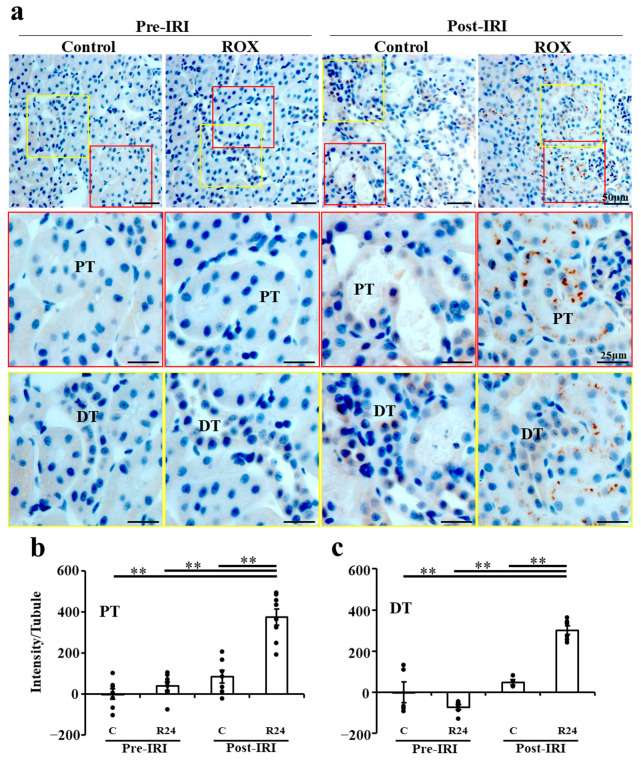
Immunohistochemical study of Epo production sites by kidney in IRI. (**a**). Epo staining was not observed in control and ROX without IRI. Epo staining is detected in 24 h ROX but not in control after IRI (upper panel). Strong vesicular staining was detected only in the proximal tubules (middle panel) and weak staining was detected in the distal tubules (distal convoluted tubules and cortical collecting ducts) (lower panel) after IRI with 24 h ROX. (**b**,**c**). DAB staining intensity of proximal tubules and distal tubules. ** *p* < 0.001 by ANOVA and Scheffe’s multiple comparison, n = 4–8. PT, proximal tubule; DT, distal tubule.

**Figure 6 ijms-25-01223-f006:**
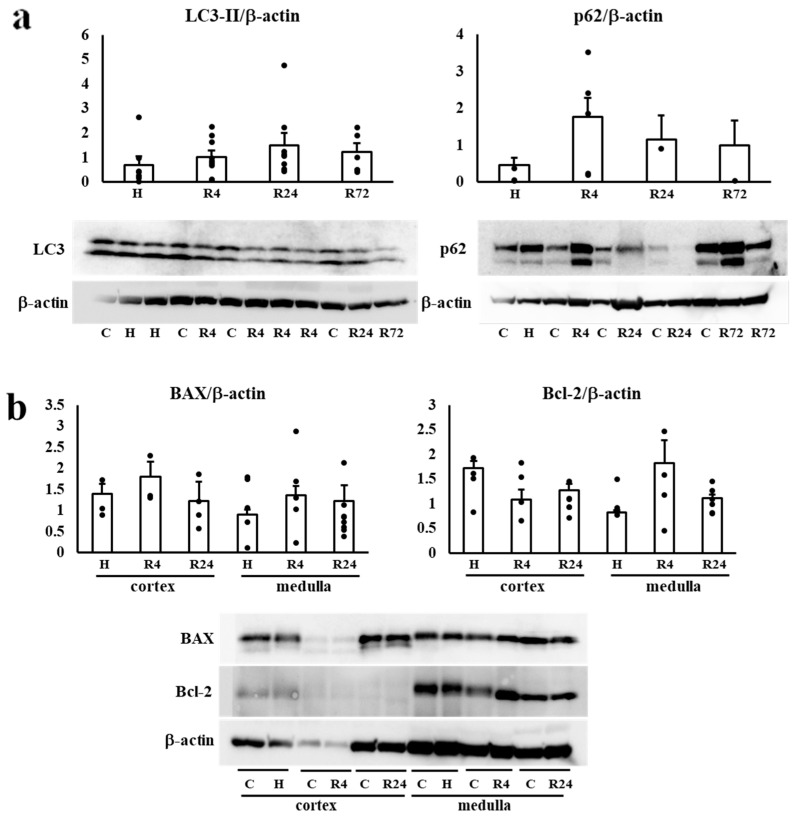
(**a**). Effects of hypoxia and ROX on autophagy. The effect of hypoxia and ROX on autophagy was examined. LC3-II (lower bands) and p62 (upper bands) expression was not changed by hypoxia and ROX. n = 4–17. C, control; H, hypoxia; R, ROX. (**b**). Effects of hypoxia and ROX on apoptosis. The effect of hypoxia and ROX on BAX and Bcl-2 expression. Hypoxia and ROX did not change BAX and Bcl-2 expression in renal cortex and medulla. C, control; H, hypoxia; R, ROX.

**Figure 7 ijms-25-01223-f007:**
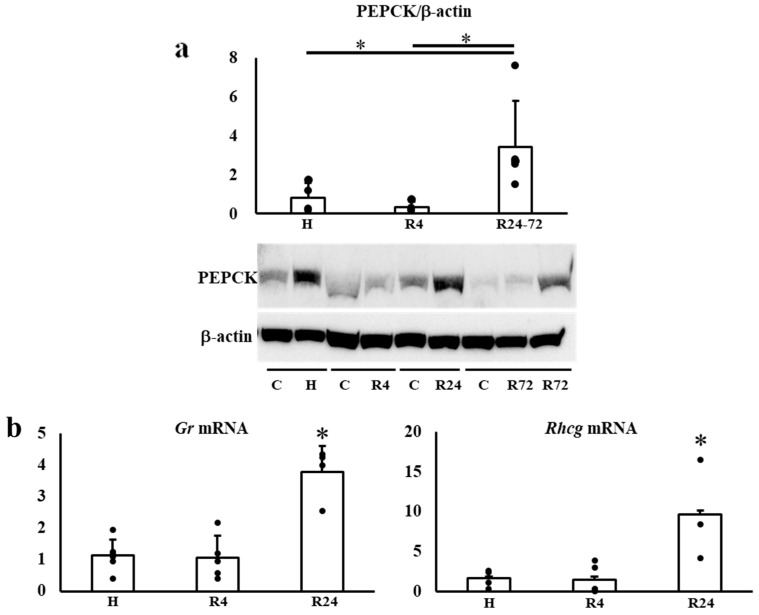
(**a**). Effects of hypoxia and ROX on nephron-specific protein expression after IRI. PEPCK expression was increased by R24-72. * *p* < 0.05 by ANOVA and Dunnett’s multiple comparison, n = 4–5. C, control; H, hypoxia; R, ROX. (**b**). Effects of hypoxia and ROX on nephron-specific mRNA expressions after IRI. *Gr* (*Nr3c1*) mRNA and *Rhcg* mRNA expressions were increased in R24 after IRI. * *p* < 0.05 by ANOVA and Scheffe’s multiple comparison. n = 3–7. C, control; H, hypoxia; R, ROX.

**Table 1 ijms-25-01223-t001:** The mRNA expression of *Hif2α*, *Hif1α*, and *Phd2* in the renal cortex before and after IRI. The mRNA expression of *Hif2α*, *Hif1α*, and *Phd2* was examined in the renal cortex before and after IRI. The decrease in *Phd2* mRNA expression was observed in the R24 and R72 groups after IRI. Data are expressed as mean ± SD. n = 3–7. * *p* < 0.05 by Student’s *t*-test.

		*HIF2α*	*HIF1α*	*PHD2*
Pre-IRI	H	1.16 ± 0.16	0.81 ± 0.11	0.91 ± 0.52
	R4	0.71 ± 0.74	9.37 ± 11.66	0.91 ± 0.53
	R24	1.27 ± 0.53	2.81 ± 3.28	1.01 ± 0.82
	R72	0.91 ± 0.64	0.56 ± 0.02	1.26 ± 1.05
Post-IRI	H	1.12 ± 0.24	1.14 ± 0.82	0.76 ± 0.63
	R4	1.25 ± 0.91	0.77 ± 0.47	1.65 ± 1.72
	R24	1.04 ± 0.23	1.54 ± 1.87	0.54 ± 0.29 *
	R72	1.20 ± 0.91	1.05 ± 0.21	0.48 ± 0.30 *

## Data Availability

The data presented in this study are openly available. Samples of the compounds are available from the authors.

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
