# Peer review of "Tubular Endogenous Erythropoietin Protects Renal Function against Ischemic Reperfusion Injury"

_ijms, 2024, doi:10.3390/ijms25021223_

Round 1

Reviewer 1 Report

Comments and Suggestions for Authors

The manuscript is well organized and the work is comprehensively described. Unufortunately, in my version of the manuscript, Figures 9 and 10 and Table 1 are not visible, while the text below them is. Please attach the above.

It would be convenient to mention where the authors see the potential application of the research results in everyday human medicine, that is, what are the potential perspectives of these results in human medicine.

Minor comments.

The unit of measurement mg/dl is correctly written as mg/dL.

Line 105. the space is missing between the word and number ("...compared with control, Figure5")

Line 227 typo - "productio of tubular"

Author Response

Answers to the Reviewer 1

    We completely agree with Reviewer 1’s comments.  We changed our manuscript according to your suggestions.

Major comments

Missing Figures and Table

     We are very sorry for the missing of Figures 9 and 10 and Table 1.  We added Figures 9 and 10 and Table 1.  Number of figures were changed.

Potential application of the research results in everyday human medicine.

   The goal of this study is to know whether endogenous Epo production has renoprotective effects or not. We found that peritubular Epo production by the interstitial cells has no renoprotective effects as shown by the use of ESAs. We also found that tubular production by proximal and distal tubules has renoprotective effects in IRI. Our model is close to the situation of kidney transplantation. The use of PHD inhibitors 24 hrs before transplantation may largely decrease the damage after transplantation. Our results also suggest that the drugs or therapy which increase tubular Epo production may suppress the decline of renal function in CKD. We and others have shown that the renin-angiotensin-aldosterone system (RAS) regulates tubular Epo production, suggesting that the drugs regulating RAS may have the potential to reduce the decline of renal function in patients with CKD. We should try to find a therapy which induces tubular Epo production and reduces the decline of renal function.  Some of this description was added to the discussion.

Minor comment

   We changed mg/dl to mg/dL.

   We put the space and corrected the misspellings.

The list of changes

  1. Figures 1-2: The standard error of the mean was added.
  2. Figures 3-5 were numbered as new Figures 3a-c. Figures 6, 7 and 8 were changed to new Figures 4, 5 and 6, respectively. Figures 9 and 10 were combined as new Figures 7. Bar graphs of Figures 5-10 were changed to bar chart plus scattered chart.
  3. Missing Figures 9 and 10 and Table 1 were recovered.
  4. The images of gels in Figures 3-5 were changed according to the comments by Reviewer 2.
  5. The new bar chart plus scattered chart graph was added to new Figure 5 to show the statistical significance.
  6. References 25, 26 and 55 were added.
  7. Extensive editing of English language was performed by Dr. Prof. Jeff Sands.

We thank Reviewer 1 for his (or her) useful comments.

Reviewer 2 Report

Comments and Suggestions for Authors

Overview

The article attempts to explore the mechanism of tubular endogenous EPO in protecting renal function from ischemia-reperfusion injury. However, the research design was unreasonable, using only one grouping mode to observe changes in some molecular biological indicators of animals. Without rescue experiments, it was impossible to determine whether these changes were related or causal. Moreover, the key indicators of immunoblotting, protein blotting, and immunohistochemical staining in the entire experimental results were of low quality, and the current research results cannot be obtained. Therefore, we do not recommend publication.

Details

1. The points in the scatter plot cannot see the density clearly due to the use of solid points. Please modify them all to hollow circles. Meanwhile, these graphs do not reflect the characteristics of distribution and require a combination of scatter plots + bar charts to reflect the mean and standard deviation/standard error of the data.

2. Most Western Blotting images are overexposed, making it difficult to distinguish between specific intensities and bands. We believe that the author should choose representative images with higher quality for these blotting images.

3. Due to the excessively high background and complex analysis of the immunoblotting area used by the author in the experiment, the author needs to accurately describe the selected analysis area for semi quantitative analysis of the blotting images and explain the specific principles of removing background and conducting relative quantitative analysis. If necessary, accompanying figures and data tables should be provided.

4. The quality of immunohistochemical staining images of the kidneys is not high, and semi quantitative analysis and statistical graphs should be provided instead of only representative images. Currently, it appears that the pathological differences between representative images are not significant.

5. LC3 is divided into LC3A and LC3B. So which part determine the author’s semi-quantitative testing, based on which basis? These all need to be carefully explained. Moreover, please clarify which band in the blotting image is responsible for p62.

6. The author used two types of internal references, β- Actin and α- Tubulin, what is the reason? Has it had an impact on the semi quantitative analysis of the article.

7. By convention, all sequences should be capitalized + all letters in italics, for example, GAPDH should be Gapdh, thus, the author is requested to modify the unprofessional presentation of the whole text.

8. For the sake of data transparency, all bar charts should be changed to bar chart plus scatter chart format.

9. The image layout seems to lack logic. The results of detecting different indicators for the same group can be grouped together to form a combined figure. Please adjust the figures to only 6-8 combined figures.

10. The main flaw of the article is that the design is too simplistic, using the same grouping pattern to detect different indicators with no rescue experiment on relevant pathophysiological or molecular biology indicators, so it is impossible to determine whether the observation results are related or causal.

11. The article did not test the success indicators of IRI modeling, including renal function, urinary protein, etc. Pathological tissue sections cannot prove changes in renal function.

Comments on the Quality of English Language

Extensive editing of English language required.

Author Response

Answers to the Reviewer 2

    We agree with the suggestions by the Reviewer 2. We changed our manuscript according to your suggestions.

Major comments

   We examined kidney injury by IRI by measuring serum creatinine, plasma Epo concentrations, mRNA and protein expression in the kidney. Our IRI method is a standard method to introduce IRI in kidney. We observed the changes of the color of the kidney by clamping the renal pedicle. Since the rats used were not so big, it was not easy to collect the urine. Most reports (Zhang M, et al, Clin Exp Pharmacol Physiol, 2021 Ref. 26, Miao A-F, et al. Renal Failure 2021、Ref. 25) investigated the changes of serum creatinine/BUN and kidney HE staining to ensure the occurrence of IRI. You commented that pathological tissue sections cannot provide changes in renal function. It is true, however, HE staining is one of the usual way to know the sites of kidney injury. Therefore, we added the HE staining of kidney to show the kidney injury together with the changes in serum creatinine levels. Urinary protein excretion does not correlate with the decrease of renal function but reflects the damage to the glomerulus. IRI is known to cause damage mainly in the proximal tubules. We think that our results clearly show the production of IRI by our methods.

Specific comments

  1. We added the standard error of the mean to Figures 1 and 2.
  2. We changed Western blotting images of Figures 3-5 (New Figures 3a-c). We measured deglycosylated Epo production at 22 kDa, which is more accurate than the glycosylated Epo bands at 34-43 kDa. Therefore, the band intensity was previously adjusted to see the bands at 22 kDa. However, we changed the images to clearly see both glycosylated and deglycosylated Epo.
  3. Since we measured the deglycosylated Epo bands at 22 kDa, the size of 22 kDa was clearly indicated the arrowhead in Figures 3-5 (New Figures 3a-c). Recombinant rat Epo was used as the standard in Figures 3-5 (New Figures 3a-c).
  4. We changed the images of immunohistochemistry with more high quality images. The expression of brown dots (Epo) was measured by an image analysing system (ZEN 2, Carl Zeiss) and was statistically significant differences were examined by ANOVA and Scheffe’s multiple comparison.
  5. Since LC3-II (B) represents autophagy, we measured the expression of LC3-II according to the method by Mizushima N, et al. (Autophagy 2007, Ref. 55). The p62 band was indicated as upper bands in new Figure 6b.
  6. We used two types of internal references for Western blotting. There are no differences between -actin (42 kDa) and -tubulin (50 kDa). We measured the expression of internal standards after stripping and reprobing the membrane. Since PEPCK (66 kDa) is much larger than deglycisylated Epo (22 kDa), we prefer to use a-tubulin for the internal reference.
  7. The sequences were changed to capitalized + italics according to your suggestions.
  8. We changed all bar charts to bar chart plus scatter chart except Figures 3a and 3b, which are typical presentation of one gel.
  9. The figures were grouped to 7 figures.
  10. We thank you for your comments on simplicity.
  11. As we mentioned above, we used the standard method to induce IRI in rats. Our data in serum creatinine levels and histology clearly indicated the success of making IRI. Measuring urinary protein is important, but it does not correlate with renal function. Renal protein excretion reflects the degree of glomerular damages, which is not seen in usual IRI models. As you mentioned, urinary protein excretion increases with the progression of renal failure in patients with diabetic nephropathy. But urinary protein excretion is low in patients with hypertensive nephropathy (benign nephrosclerosis). Since protein intake affects BUN, serum creatinine is the best indicator of renal function.
  12. References 25, 26 and 55 were added.

Extensive editing of English language was performed by Dr. Prof. Jeff Sands.

The list of changes

  1. Figures 1-2: The standard error of the mean was added.
  2. Figures 3-5 were numbered as new Figures 3a-c. Figures 6, 7 and 8 were changed to new Figures 4, 5 and 6, respectively. Figures 9 and 10 were combined as new Figures 7. Bar graphs of Figures 5-10 were changed to bar chart plus scattered chart.
  3. Missing Figures 9 and 10 and Table 1 were recovered.
  4. The images of gels in Figures 3-5 were changed according to the comments by Reviewer 2.
  5. The new bar chart plus scattered chart graph was added to new Figure 5 to show the statistical significance.
  6. References 25, 26 and 55 were added.
  7. Extensive editing of English language was performed by Dr. Prof. Jeff Sands.

We thank Reviewer 2 for his (or her) useful comments.

Round 2

Reviewer 2 Report

Comments and Suggestions for Authors

Overview

The authors spent several days simply answering our questions, but in fact, there was no essential progress in the article. Moreover, we found that the author's team had a serious lack of understanding of kidney pathophysiology, and we have serious doubts about the authenticity and repeatability of the research results, so we do not recommend publication. Specific research comments are evaluated as follows:

Responses to the author's major comments

We do not deny the need for HE staining, we emphasize that HE staining does not reflect changes in kidney function, and kidney function is not only the so-called creatinine and BUN changes that the authors answered, and there are renal injuries that do not even change. The authors claim to study kidney injury without detecting or even mentioning functional proteins that reflect proximal tubule injury, urinary proteins, and the most basic eGFR estimates of CCr. We believe that the authors' study in this area lacks the most basic pathophysiological basis.

Detail comments

1. The author still did not change the image of the value, the image is still solid points, rather than hollow points, it is impossible to see each data clearly. Besides, some of the images are not adjusted into scatter charts with only bar charts.

2. The author claimed to have modified the WB blotting image, but in fact it was still the original image with overexposure and poor quality. We seriously doubt whether the author could not repeat his experimental results or provide more clear data, which indicates that there are major problems in the author's research method and research stability.

3. The author admits that two kinds of internal parameters are used, but does not put forward the theoretical and experimental basis that there is no difference between the two kinds of internal parameters.

4. The author thanked us for our comments on the simplicity of the article, but did not make any changes to the article in this regard, so far our comments remain the same:

The main flaw of the article is that the design is too simplistic, using the same grouping pattern to detect different indicators with no rescue experiment on relevant pathophysiological or molecular biology indicators, so it is impossible to determine whether the observation results are related or causal.

If the author is unable to resolve this issue, we do not recognize the reliability of the article and do not recommend publication.

5. The authors claim that much of the editing of the English language was done by Prof. Dr. Jeff Sands. The author is requested to provide the corresponding editorial proof as well as the professor's research institute and previous research paper for reference whether he/she is eligible for revision.

Comments on the Quality of English Language

The authors claim that much of the editing of the English language was done by Prof. Dr. Jeff Sands. The author is requested to provide the corresponding editorial proof as well as the professor's research institute and previous research paper for reference whether he/she is eligible for revision.

Author Response

Answers to Reviewer 2

    We thank Reviewer 2 for useful comments.

     You requested that we examine eGFR or Ccr. The formula to calculate eGFR from serum creatinine is different according to the country. eGFR is used to compare the renal function with different height, body weight, gender and age. But the rats we used have almost the same body weight and are all male. Serum creatinine should show the same change as eGFR. We know the formula to calculate human eGFR, but we do not know the formula to calculate rat eGFR. Urine collection is required to calculate Ccr. A large amount of water intake by tube or intravenous injection of saline, and the use of metabolic cages, are required to measure Ccr. But such procedures will give a lot of stress to the rats. The rats are suffering from IRI. To add more stress to the recovering rats should be avoided since we can measure serum creatinine. Such procedures should not be planned from the point of animal care and protection. Our experimental animal committee or ethic committee will not permit such experiments for the IRI rats.

  1. Figures 3 and 4 are a typical gel of the Western blotting as we described in the figure legends. Therefore, Figures 3 and 4 were not changed to the scatter charts.
  2. We changed Figure 3 to the new gel.
  3. We changed the internal standard in Figure 7 from α-tubulin to β-actin according to your comments. Therefore, statistical significance was slightly changed.
  4. We thank you for your comments. We measured Epo protein and mRNA expression. Our deglycosylation-coupled Western blotting could provide novel data. No other groups have investigated Epo using our methods.  We did not measure the HIF2α and HIF1α protein expressions in the kidney since there are no reliable antibodies against them.
  5. This manuscript was again checked by Professor Dr. Jeff M. Sands. He is the Juha P. Kokko Professor of Medicine and Renal Division Director at Emory University in Atlanta, GA, USA. He was the Editor-in-Chief of the American Journal of Physiology – Renal Physiology from 2001-2007 and the 91st President of the American Physiological Society from 2018-2019. He received Homer W. Smith Award in 2022, which is the most prestigious scientific award presented by the American Society of Nephrology. He has published more than 175 peer-reviewed papers, over 100 invited reviews/book chapters, and co-edited 1 book.

Klein JD, Khanna I, Pillarisetti R, Hagan RA, LaRocque LM, Rodriguez EL, Sands JM. An AMPK activator as a therapeutic option for congenital nephrogenic diabetes insipidus. JCI Insight 6: e146419, 2021. 

https://doi.org/10.1172/jci.insight.146419 PMCID: PMC8119225 

Kuma A, Wang XH, Klein JD, Tan L, Naqvi N, Rianto F, Huang Y, Yu M, Sands JM. Inhibition of urea transporter ameliorates uremic cardiomyopathy in chronic kidney disease. FASEB J. 34: 8296-8309, 2020. https://doi.org/10.1096/fj.202000214RR PMCID: PMC7302978 

List of changes

  1. Figure 1 was changed to a new gel.
  2. Internal control of Figure 7a was changed from a-tubulin to b-actin. Therefore, scatter chart/bar graph and statistical significance was slightly changed. We thank Reviewer for the comments.
  3. The manuscript was checked by Professor Dr. Jeff M. Sands.

Round 3

Reviewer 2 Report

Comments and Suggestions for Authors

The authors' team studying renal physiology but did not know the basic formula for estimating glomerular filtration rate using creatinine, and functional proteins and urinary proteins with proximal tubule injury remained unanswered. We recommend the authors to change to scatter plots because of the principle of data transparency, and it seems that the authors did not change all the images. In addition, the quality of the new gel is still not high, the author modified the internal parameters but did not explain the basis of the original internal parameters. Most importantly, there has been no substantial improvement in the major deficiencies mentioned in comment 4.

Based on the above principles, we believe that this article is not suitable for publication.

Comments on the Quality of English Language

Extensive editing of English language required.

Author Response

 Answers to the Reviewer 2

    We thank the Reviewer 2 for his (or her) comments.

    The measurement of GFR using serum creatinine is important. However, our rats suffered from IRI and urine volume is small. To measure GFR correctly, the collection of urine is necessary. To collect small volume of urine, the use of metabolic cage is required. It will influence the recovery from IRI and the rats will feel pain. Such severe protocol has to be checked and has to be approved by animal committees and they will not permit the protocol from the view of animal care. We agree that urinary protein is important factor for evaluating renal function. However, many reports have investigated IRI without measuring urinary protein excretion.

   We revised Figure 3a and 3b from black to white bars. The figure legend of Figure 3 was revised. Therefore, all figures 1-7 were changed according to your suggestions.

      There are no changes in the expression of internal standards between β-actin and α-tubulin.

List of changes

  • The Figure 3a and the figure legend were revised. The gel of β-actin in Figure 3 was changed. The presentation of R24, H and R4 from left to right in the upper Figure 3a were changed to H, R4 and R24. R24, H and R24 in the right part of the Epo gel was also changed to H, R4 and R24.
  • The black bars in the Figures 3a and 3b were changed to the white bars.